# Autophagy Controls Sulphur Metabolism in the Rosette Leaves of Arabidopsis and Facilitates S Remobilization to the Seeds

**DOI:** 10.3390/cells9020332

**Published:** 2020-01-31

**Authors:** Aurélia Lornac, Marien Havé, Fabien Chardon, Fabienne Soulay, Gilles Clément, Jean-Christophe Avice, Céline Masclaux-Daubresse

**Affiliations:** 1UCBN, UMR INRA–UCBN 950 Ecophysiologie Végétale &, Agronomie & Nutritions N.C.S., SFR Normandie Végétal (FED 4277), Normandie Université, F-14032 Caen, France; 2Institut Jean-Pierre Bourgin, INRAE, AgroParisTech, Université Paris-Saclay, 78000 Versailles, France

**Keywords:** sulphate, leaf senescence, seed filling, nitrogen use efficiency, sulphur use efficiency, resource allocation

## Abstract

Sulphur deficiency in crops became an agricultural concern several decades ago, due to the decrease of S deposition and the atmospheric sulphur dioxide emissions released by industrial plants. Autophagy, which is a conserved mechanism for nutrient recycling in eukaryotes, is involved in nitrogen, iron, zinc and manganese remobilizations from the rosette to the seeds in *Arabidopsis thaliana*. Here, we have compared the role of autophagy in sulphur and nitrogen management at the whole plant level, performing concurrent labelling with ^34^S and ^15^N isotopes on *atg5* mutants and control lines. We show that both ^34^S and ^15^N remobilizations from the rosette to the seeds are impaired in the *atg5* mutants irrespective of salicylic acid accumulation and of sulphur nutrition. The comparison in each genotype of the partitions of ^15^N and ^34^S in the seeds (as % of the whole plant) indicates that the remobilization of ^34^S to the seeds was twice more efficient than that of ^15^N in both autophagy mutants and control lines under high S conditions, and also in control lines under low S conditions. This was different in the autophagy mutants grown under low S conditions. Under low S, the partition of ^34^S to their seeds was indeed not twice as high but similar to that of ^15^N. Such discrepancy shows that when sulphate availability is scarce, autophagy mutants display stronger defects for ^34^S remobilization relative to ^15^N remobilization than under high S conditions. It suggests, moreover, that autophagy mainly affects the transport of N-poor S-containing molecules and possibly sulphate.

## 1. Introduction

Macroautophagy consists in the formation of cytosolic double membrane vesicles named autophagosomes that engulf and sequester unwanted cytoplasmic constituents to drive them to the central vacuole. The vacuole hydrolases and proteases then degrade unwanted material and release amino acids and sugars to be recycled [1]. In plants, macroautophagy is enhanced in senescing leaves. By contributing to the degradation of useless cytoplasmic components and altered organelles, it facilitates the release of nutrients that are then available to the source for sink mobilization [2,3]. In Arabidopsis and maize, it was shown using mutants that functional autophagy is essential for the remobilization of nitrogen from vegetative organs to the seeds [4,5]. More recently, Chen et al. [6] showed that enhancing autophagy in Arabidopsis by overexpressing *ATG8* genes leads to a better N remobilization to the seeds.

Autophagosome formation incorporates the products of many *AUTOPHAGY* (*ATG*) genes, among which eighteen are essential for the formation of autophagosomes [7]. The *ATG5* gene is one of these core-machinery genes. Like other autophagy mutants, the *atg5* mutants display a hypersenescent phenotype, having accumulated a large amount of proteins in their rosettes during ageing [5,8]. The ^15^N labelling experiments showed that N remobilization to the seeds was decreased by 40% in the *atg5* mutants relative to the wild type, and the combination of *atg5* mutations with salicylic acid (SA) deficient mutations partially abolished the senescence phenotypes but not the N remobilization deficiency. Recently, it was shown that overexpressing *ATG5* in Arabidopsis stimulated autophagosome formation, autophagic flux, delayed senescence and increased resistance to necrotrophic pathogen, plant growth, seed set and seed oil content [9].

Because the cargoes taken in charge by macroautophagy for degradation are not only proteins but also organelle fragments, autophagy is certainly not only involved in the recycling of proteins or nitrogen sources. Recent advances show that it can also contribute to the recycling of lipids and of any kind of micronutrients contained in the macromolecules [3,10,11]. Accordingly, it was recently shown, using ^57^Fe labelling and measuring metal budgets, that the efficiencies of iron, zinc and manganese translocation from vegetative organs to the seeds were dramatically decreased in the *atg5* mutant [10].

Like nitrogen, sulphur (S) is an essential element for growth and metabolic functioning in plants [12]. It is an important constituent of proteins due to S-containing amino acids like methionine and cysteine. S fertilization in crops has been an agricultural concern since the 1980s. Due to the decrease of atmospheric sulphur dioxide (SO_2_) from industrial emissions, S deposition into the soil was strongly reduced, leading to the increasing occurrence of S deficiency in crops, at least in Western Europe [13,14]. *Brassicaceae* plants are highly S-demanding, as they have high concentrations of S-containing secondary metabolites, and S limitation can severely impact their seed yield and quality [15]. Therefore, the number of studies on S use efficiency started increasing since the 1980s.

In this study we used the same genotypes as Guiboileau et al. [5] to investigate the role of autophagy on sulphur remobilization to the seeds. Simultaneous labelling with ^34^SO_4_^2−^ and ^15^NO_3_^−^ was performed in order to compare the N remobilization efficiency (NRE) and the S remobilization efficiency (SRE). Our results indicate that SRE was surprisingly higher under high sulphur conditions than under sulphur limitation in both autophagy mutants and control lines. However, the *atg5* mutants were less efficient in S remobilization than the control lines under both conditions. In addition, by comparison with N remobilization, we show that low S conditions exacerbate the defects of autophagy mutants for S remobilization, suggesting the occurrence of a selectivity with regard to autophagy in the different S sources available in plants for remobilization.

## 2. Materials and Methods

### 2.1. Plant Material and Growth Conditions

Arabidopsis mutants *atg5-1* [SALK_020601], *atg5-2* [SAIL_129B07], *sid2.atg5-1* and NahG*.atg5-2.* were the same as used before by Guiboileau et al. [5]. The plants were grown on sand on either high sulphate supply (High S; 0.266 mM SO_4_^2−^, 10 mM NO_3_^−^) or low sulphate (Low S; 0.016 mM SO_4_^2−^, 10 mM NO_3_^−^) conditions. For the S and N remobilization experiment, the plants were sown on sand and cultivated under high S conditions for four weeks in short days (8 h light–16 h dark, 150 µmoL/m^2^/s). Then, the plants were labelled for five days using a labelled high S nutrient solution containing ^15^N and ^34^S isotopes (^15^N 10% enrichment and ^34^S 2.5%). After 5 d labelling, some of the plants were harvested for isotopic ratio measurement, to estimate the level of ^34^S and ^15^N uptake over five days by the different genotypes. The pots containing sand and roots of the remaining plants, dedicated to measure ^34^S and ^15^N remobilizations, were rinsed carefully several times with deionised water. Then, half of these plants were transferred to unlabelled low S conditions and half to unlabelled high S conditions. The plants were then grown in short days (8 h light) for eight weeks before the transfer to long days (16 h light). Similar day/night temperatures (21 °C day, 17 °C night) were maintained until seed maturity.

### 2.2. ^15^N Labelling and Tracing

The plants were harvested at the end of their reproduction cycle, when both vegetative tissues and seeds were dry. At harvest, the plants were divided into the following parts: rosette, stem (including cauline leaves), silique envelopes (pericarp) and seeds. The dry weight (DW) of all these compartments was determined. Then, the ^34^S and ^15^N abundances in all collected samples were determined as described in Diaz et al. [16]. The ^34^S abundance was calculated as atom percent and defined as A(^34^S)% = [100 × (^34^S)/(^34^S + ^35^S)] and the ^15^N abundance as A(^15^N)% = [100 × (^15^N)/(^15^N + ^14^N)]. The S and N concentrations (S% and N%) as mg of element per 100 mg of dry weight were also determined. The enrichment (E%) was calculated as (Asample% – Acontrol%). Acontrol% was determined for unlabelled samples. The absolute quantity of ^34^S contained in a sample was defined as [DW × E(^34^S)% × S%] and the absolute quantity of ^15^N contained in a sample as [DW × E(^15^N)% × N%]. The partitioning of ^34^S was calculated for each organ (rosette, stem, pericarp and seeds) as the proportion of the quantity of ^34^S in each organ relative to the total quantity in the whole plant, and expressed as % of the whole plant. As such, the sum of the partitionings (rosette, stem, pericarp and seeds) was 100%. The same was done for the ^15^N partitioning. Unlabelled samples were harvested in order to determine the natural abundance of ^34^S and ^15^N.

### 2.3. Metabolite Profiling and Sulphate Measurements

Metabolite profiling and metabolomics data processing were performed according to Masclaux-Daubresse et al. [17]. The plant material used for the metabolomics analyses consisted in the rosette of 60-days-old plants grown under high S or low S conditions and short days. Sulphate concentrations were measured on the dry matter obtained from the ^15^N labelled plants at seed maturity. Sulphate was extracted from ca. 25 mg of freeze dried matter (DM), which was initially mixed up with 1 mL of 50% ethanol [18]. After incubation at 45 °C for 1 h, the extract was centrifuged at 10,000g for 10 min, and the supernatant was collected. The ethanol extraction step was reiterated on the pellet, and the supernatant was pooled with the previous one. The pellet was then re-extracted in 1 mL of distilled water at 95 °C for 1 h. The extracts were centrifuged at 10,000g for 10 min. All the supernatants were pooled and evaporated under vacuum (Concentrator Evaporator RC 10.22, Jouan, Saint-Herblain, France). The dry tissue was re-suspended in 0.5 mL of ultrapure water and filtered to remove any plant residues. The sulphate concentration was determined by high performance liquid chromatography (HPLC, DX100, Dionex Corp., Sunnyvale, CA, USA). The eluent solution consisted of 1.8 mM Na_2_CO_3_ and 1.7 mM Na_2_HCO_3_, and was pumped isocratically over an AS17 guard column.

### 2.4. Statistical Analysis

The ^34^S and ^15^N labelling experiments were carried out in two independent biological repeats (R1 and R2), performed as two consecutive culture cycles in the same growth chamber. An ANOVA Newman–Keuls (SNK) comparison was performed using XLSTAT (13.2). As R1 and R2 provided very similar data, the mean and SD were calculated based on the R1 and R2 data (n = 12; 6 plants per genotype for each repeat, R1 and R2). For metabolomics, an ANOVA was first used to select the significant metabolites. The subset of significant metabolites was then used to perform hierarchical clustering using MEV_4_8_1 free software (https://sourceforge.net/projects/mev-tm4/).

## 3. Results

### 3.1. Hypersensitivity of Autophagy Mutants to Sulphur Limitation

It is well known that autophagy mutants are hypersensitive to carbon and nitrogen starvations [2]. When submitted to chronic sulphur limitation, *atg5* mutants displayed earlier leaf yellowing phenotypes (Figure 1A), which suggested defects in the management of sulphur resources.

It is also well known that autophagy mutants accumulate salicylic acid. The combination of *atg* mutations with *sid2* mutation or with NahG overexpression has already been used by Guiboileau et al. [5] to investigate the SA-independent autophagy effects on N remobilization. We have thus used the same genotypes as Guiboileau et al. to determine how biomass partitioning and sulphur allocation are modified in plants defective in autophagy, independently of the salicylic acid effect. Nitrogen and sulphur labelling were performed on four-week-old plants providing ^34^SO_4_^2−^ and ^15^NO_3_^−^ every day during five days (Figure 1B). Since N remobilization has already been monitored by Guiboileau et al. [5], we performed double labelling with ^34^S and ^15^N in order to use ^15^N as a control of the source to sink nutrient fluxes. After labelling, plants were watered with isotope-free solutions and grown under low or high sulphur conditions until seed maturity. Under both low and high sulphur conditions, the *atg5-1*, *atg5-2*, *sid2*.*atg5-1* and *NahG*.*atg5-2* plants were (64% to 78%) smaller at harvest than their respective control lines (i.e., Col-0 for *atg5-1* and *atg5-2,* sid2 for *sid2.atg5-1*, and NahG for NahG*.atg5-2*; Figure 1C). However, the decrease of plant biomass in the *atg5* mutants relative to the control lines was globally similar under low S and high S conditions.

Histograms representing the partition of dry weight (DW) in the rosettes, stem, pericarp and seeds (Figure 2) show that the patterns were similar under low S and high S for all the genotypes. Under both low S and high S, the partition of the biomass in the rosettes of *atg5* mutants was more important relative to the control lines, and the biomass allocated to siliques (both pericarps and seeds) was lower in the *atg5* mutants than in the control lines. This was consistent with the lower harvest index already reported by Guiboileau et al. for autophagy mutants [5]. It can be noticed that the partition of DW in the stems was poorly modified in the *atg5* mutants by comparison with the control lines (less than 10%), irrespective of S conditions (Fig2A–B). Thus, it showed that autophagy mainly impacted the rosette and seed parts.

### 3.2. Autophagy Mutants Hyper-Accumulate Sulphur

As expected, total sulphur concentrations (S%; mg.100 mg^-1^ DW) were lower in the plants grown under low S than in those grown under high S (Figure 3). In the control lines, S% was higher in the seeds than in the other organs, irrespective of S conditions. This indicated that a large part of the S resources in plants was allocated to the seeds at harvest in the control lines. This feature was in clear contrast with that of *atg5* mutants, whose S% in the rosette, stem and pericarp (i.e., dry remains) remained almost as high as the S% measured in their seeds. It is noticeable that the S% in all the organs of *atg5* mutants were higher than those measured in the control lines, except in seeds under high S.

The reason why sulphur concentrations were higher in the *atg5* plants than in controls is certainly related to their lower growth rate, that resulted in lower S dilution. Nevertheless, the strong accumulation of S, in their dry remains especially, suggests that autophagy mutants displayed defects in S reallocation during plant development.

Although the genotype effects on the S% in the rosette, stem and pericarp were similar under low S and high S, we can observe that the genotype effect on the S% in seeds is different depending on high S and low S conditions. The S concentration in seeds was lower in the control lines than in *atg5* mutants under low S, but similar under high S. This reveals that S limitation was more critical in the control lines than in *atg5* mutants under low S. Again, this could be explained by the fact that *atg5* plants were smaller than control plants.

The values of S% measured in seeds raise two questions: (i) How can the S% (S concentration) in the seeds of the *atg5* mutants be maintained as high as in the control lines, while S remobilization to the seeds is lower in the *atg5* mutants relative to the control lines? (ii) Why is the S% in the seeds of *atg5* similar to that of the control lines, while the S% in the other organs are much lower in *atg5* mutants than in the control lines? The explanation of this intriguing observation is that the lower S flux in autophagy mutants led to (i) a lower amount of S allocated to the reproductive organs and (ii) less seeds produced [19]. Indeed, the biomass of the rosette (source leaves) was 2 times lower in the *atg5* mutants than in the control lines at harvest, irrespective of S conditions. The biomass of the seeds of the *atg5* mutants was 4.5 and 6.2 times lower than that of the control lines under low S and high S, respectively. The S content (in mg) in the *atg5* mutants was 3 and 6 times lower than that of the controls under low S and high S, respectively. The ^34^S content (in mg) in *atg5* mutants was 3.5 and 3.2 times lower than that of the control lines under low S and high S, respectively. Therefore, we see that the absence of autophagy in Arabidopsis had a stronger effect on seed production than on the allocation of S to the seeds. As a result, the autophagy mutants produced less seeds properly filled with S and with N [5].

### 3.3. Sulphur Remobilization from the Rosette to the Seeds and Pericarp Is Less Efficient in atg5 Mutants

The ^34^S labelling was performed before transferring the plants to S limitation. The total ^34^S quantity absorbed by the different plants was globally similar, irrespective of genotypes and sulphur conditions (Appendix A). However, the ^34^S allocation at harvest was different in the control lines and *atg5* mutants (Appendix A).

The ^34^S partitioning (also called ^34^S harvest index, ^34^SHI) gives a clear picture of the ^34^S remobilization from the rosette to the stems, pericarp and seeds, independently of the differences that might exist in plant biomass (Figure 4). Surprisingly, ^34^S remobilization was more efficient under high S than under low S (Figure 4). Almost 80% and 65% of the ^34^S was remobilized from the rosettes to the rest of the plant in control plants under high S and low S, respectively. In *atg5* mutants, this amount was significantly lower, representing only 50% and 25% under high S and low S, respectively. In the control plants, the largest proportion of ^34^S was found in the seeds, irrespective of the nutritive conditions, and only 13%–12% has remained in the stems and pericarps. The proportion of ^34^S remaining in the stems of *atg5* (13% in average) was globally similar to that found in the control lines. Only the ^34^S remobilization to the seeds was significantly different between the control lines and *atg5* mutants. The allocation of ^34^S to the seeds was 5 times and 2 times lower in *atg5* than in the control lines under low S and high S, respectively (Figure 4).

### 3.4. Relative Rates of ^34^S and ^15^N Remobilizations Are Different in Control Lines and atg5 Mutants Under Low S Conditions

Under both low S and high S, the ^34^S and ^15^N remobilizations to the seeds were significantly higher in the control lines than in *atg5* mutants (Figure 5A–D). The N and S remobilization defects in *atg5* mutants were independent of SA. As in the case of ^34^S, we observed that the ^15^N remobilization to the seeds was more efficient under high S than under low S (Figure 5B,D). This revealed that the S status of the plants was important for both N and S nutrient fluxes at the whole plant level.

To compare the ^34^S and ^15^N fluxes in both S conditions, the partitionings of ^34^S (^34^SHI) and ^15^N (^15^NHI, ^15^N harvest index) in seeds were compared considering the ^34^SHI/^15^NHI ratio. Under both low S and high S conditions, the remobilization of ^34^S in the control lines was more efficient than the remobilization of ^15^N (^34^SHI/^15^NHI ratio above 150–200%) (Figure 5E). This indicated that the mobilization of S from the rosette to the seeds was probably not exclusively supported by S amino acids. Under high S, the ^34^SHI/^15^NHI ratio in *atg5* mutants was mostly similar to that of the control lines. However, under low S, this ratio was interestingly much lower in *atg5* mutants (globally equal to 100%) than in the control lines (remaining equal to 150–200%, as under high S). Such discrepancy indicated that under low S (i), ^34^S remobilization to the seeds was strongly reduced in the *atg5* mutants relative to the control lines and that (ii) under low S conditions, the S forms remobilized from the source organs to the seeds in *atg5* mutants were certainly mainly S amino acid forms. It is likely that the decrease in ^34^S remobilization in *atg5* mutants relative to control lines under low S was due to defects in mobilizing N-poor S-containing molecules.

### 3.5. Metabolite Accumulation in the Rosette Leaves of atg5 Mutants Is Exacerbated under Low S Irrespective of Salicylic Acid Production

In order to determine whether amino acids, and especially S-containing amino acids accumulated in the leaf tissues of autophagy mutants, metabolite profiling was performed using GC–MS (gaz chromatography coupled to mass spectrometry). We first compared the metabolite relative contents in *atg5-1* and Col-0 under high S and low S (Figure 6). Results showed that, relative to Col-0, most of the metabolites accumulated in *atg5-1* under both low S and high S. Only urate and threitol contents were lower in *atg5-1* than in Col-0. Surprisingly, glucose, fructose, galactose, mannose, fumarate and xylose accumulated in *atg5-1* under low S, relative to Col-0, but were lower under high S. Like other amino acids, cysteine and methionine accumulated significantly in *atg5-1*. Such accumulation was consistent with the defects observed for N and S remobilization in *atg5* mutants. To determine whether the accumulation of metabolite was salicylic acid-dependent, we then compared the metabolome of *atg5-1*, *atg5-1.sid2,* Col-0 and *sid2* under low S conditions (Figure 7). Again, we observed that, globally, metabolites were more abundant in *atg5* mutants vs. the control lines, irrespective of SA (Figure 7). The few metabolites depleted in *atg5-1* vs. Col-0 (urate, threitol, linoleic acid, spermidine, dehydroascorbate and porphine) were also depleted in *atg5-1.sid2* compared to *sid2.* This then proved that the hyperaccumulation of both C and N metabolites in *atg5-1* under low S was SA-independent. This feature was consistent with the SA-independent effect of *atg5* mutation on N and S remobilization (Figure 5A–D).

In order to determine whether autophagy could impair the translocation of inorganic S from the rosette leaves to the seeds, the concentrations of the sulphate remaining in the rosettes of plants after seed production were measured using the same plant material used for the ^34^S remobilization assay. Interestingly, although sulphate availability in the soil was limited under low S conditions, the concentrations of sulphate in the dry rosettes of the *atg5-1* and *atg5-2* mutants at harvest were twice as high as those found in the dry rosettes of the Col-0 wild type (Figure 8). Sulphate concentrations were also significantly higher in *atg5-1.*sid2 and *atg5-2.*NahG than in *sid2* and NahG, respectively. This indicates that sulphate was blocked in the rosettes of the *atg5* mutants, contributing to the defect of S remobilization to the seeds.

## 4. Discussion

There are now several lines of evidence that autophagy is involved in the remobilization of macro- and micro-elements in plants [4,5,6,10]. Using ^34^S isotopes, we show here that autophagy is controlling sulphur remobilization in a SA-independent manner and is essentially blocking the S flux from the rosette leaves to the seeds. The absence of difference between the *atg5* mutants and the control lines for ^34^S partitioning in the stem and pericarps reveals that the steps mainly controlled by autophagy are (i) the export of S forms from the source rosette leaves and (ii) the import of the S-mobilized forms into the seeds. This feature is in good accordance with the expression levels of autophagy genes, that were shown to be increased in senescing leaves and in seeds during maturation. The S forms transported from source to sink in plants are not well known. Sulphur remobilization may involve organic forms, as glutathione, glucosinolates, methiine (S-methylcysteine sulphoxide) [20] and the methionine and cysteine S amino acids, and also inorganic forms such as sulphate, at least when SO_4_^2−^ availability for plants is sufficient.

The metabolite profiling performed on autophagy mutants and control lines showed that most of the amino acids, including cysteine and methionine, accumulated in the rosettes of *atg5* mutants irrespective of S nutrition. This feature is consistent with the metabolic profiling previously reported by Masclaux-Daubresse et al. [17]. In their reports, the authors also showed that glutathione highly accumulated in the rosette leaves of autophagy mutants, possibly due to excessive oxidative stress. Glutathione storage in rosette leaves of autophagy mutants may impair S remobilization.

The proteomic analyses performed by Havé et al. [21] showed that the MTO2 (methionine overaccumulator 2; threonine synthase) protein was less abundant in *atg5* mutants than in the control lines. MTO2 was shown to lead to methionine overaccumulation. Therefore, the lower MTO2 content in *atg5* mutants might explain the accumulation of methionine in the *atg5* rosettes. Higher abundances of enzymes involved in cysteine, methionine and glucosinolate biosynthesis in *atg5* mutants irrespective of SA and of growth conditions have also been reported by Havé et al. [21]. These differences, relative to the control lines, may also explain the increase of S amino acids measured in autophagy mutants (Figure 9). Nevertheless, the defects in methionine, cysteine and glutathione translocation from the leaves to the seeds in autophagy mutants remain to be explored.

As said before, autophagy is not only involved in the recycling of proteins; it can also contribute to the degradation of lipids, and especially membrane lipids [21,22]. Its impact on lipid metabolism and turnover could contribute to impair the S mobilization from cell to cell and from source organs to sinks. How autophagy impairs the mobilization of S from lipids and proteins and whether there are specific cargoes degraded by autophagy under sulphate limitation remains to be investigated.

The lower S remobilization in autophagy mutants could also be due to a lower translocation of the sulphate itself. The ^34^S labelling did not reveal any defect in the sulphate uptake in *atg5* mutants. Accordingly, the transcriptomic and proteomic data reported by Havé et al. [17,21] did not reveal any changes in the transcript or protein levels of the sulphate transporters in the *atg5* mutants. However, the higher concentrations of sulphate measured at the end of seed maturity, in the rosettes of the *atg5* mutants grown under low S conditions, strongly suggest that sulphate was not as well assimilated in autophagy mutants as in the control lines, and that the remobilization from the rosettes to the seeds was impaired in autophagy mutants.

Still, the comparison of SRE (^34^SHI) and NRE (^15^NHI) shows that S remobilization was higher than N remobilization under high sulphate conditions (2-fold). This means that the conventional S-mobile forms for source to sink mobilization in plants (both control and *atg5* mutants) are not only N-containing molecules (like amino acids, methiine or glutathione, for example) but also N-free molecules. Interestingly, the fact that under low S, the ^34^SHI/^15^NHI ratio was lower in the *atg5* mutants (from 200% to 100%) than in the control lines, indicates that SRE was more affected than NRE by low S in the *atg5* mutants. This also indicates that some S-mobile forms were missing in *atg5* relative to the control lines. These missing S-mobile forms were certainly N-poor S-containing molecules. Although their nature remains to be determined, our results suggest that the composition of the S-mobile pools in *atg5* mutants and in control lines are different, at least under low S conditions.

In conclusion, our study shows that (i) S remobilization from the rosette to the seeds is impaired in the *atg5* mutants when compared to the control lines, irrespective of SA accumulation and S conditions; (ii) many metabolites, among which S-free amino acids, as well as methionine and cysteine, over-accumulate in the rosette leaves of the atg5 mutants; and (iii) the nature of the S-mobile molecules available for remobilization is different in the *atg5* and the control lines, at least under low S conditions. Under low S, it is likely that the absence of autophagy affects the transport of N-poor S-containing molecules and that the SA effect is partially involved in this process.

## Figures and Tables

**Figure 1 cells-09-00332-f001:**
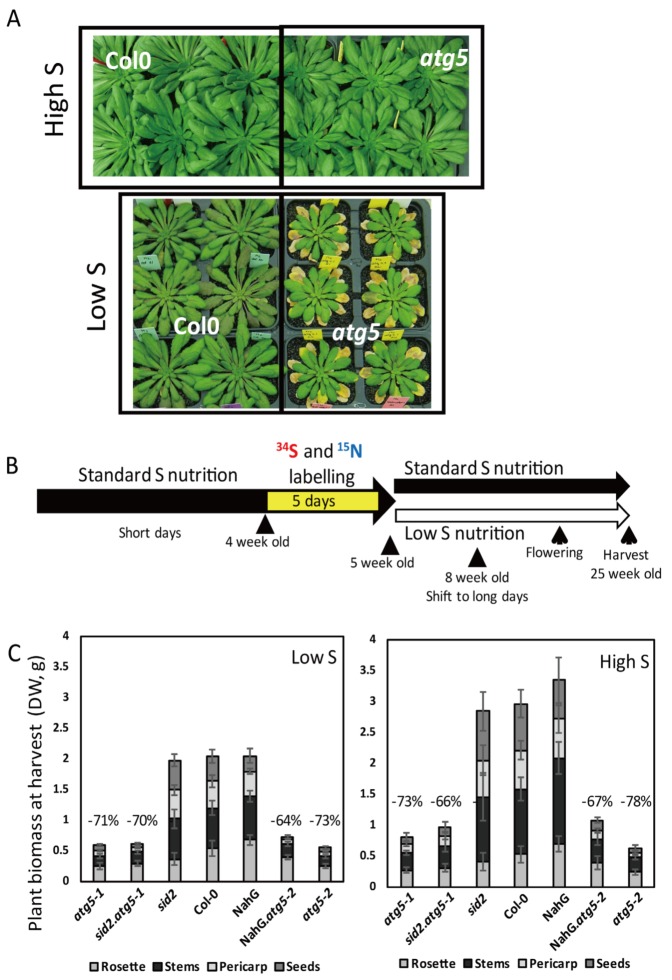
The biomass of *atg5* autophagy mutants is affected under both low sulphur (Low S) and high sulphur (High S) conditions. (**A**) Phenotype of *atg5* mutants under high S and low S conditions 60 days after sowing. (**B**) Schematic representation of the labelling experiments performed on *atg5* mutants (*atg5-1*, *atg5-2*, *atg5-1.sid2*, *atg5-2.*NahG) and control lines (Col-0, *sid2* and NahG). The plants were sown on sand and cultivated under high S conditions for four weeks in short days (8 h light–16 h dark, 150 µmoL/m^2^/s^1^). Then, the plants were labelled for five days using a high S nutrient solution containing ^15^N and ^34^S isotopes. After 5 d labelling, half of the plants were transferred to unlabelled low S conditions and half to unlabelled high S conditions. After eight weeks, the plants were transferred to long days (16 h light), maintaining similar day/night temperatures (21 °C day, 17 °C night) until seed maturity. (**C**) The biomasses of the rosette, stems, pericarps and seeds were determined at harvest on the ^34^S and ^15^N labelled plants (dry weight, DW, g). The numbers above the *atg5* mutant bars indicate the % of biomass decrease in mutants relative to their respective control lines. Data are the adjusted means and SD from two biological repeats with six plants each (n = 12). The different letters indicate values of total biomass significantly different at *p* < 0.05 (n = 12) between *atg5* mutants and control lines, as determined using an ANOVA Newman–Keuls (SNK) comparison.

**Figure 2 cells-09-00332-f002:**
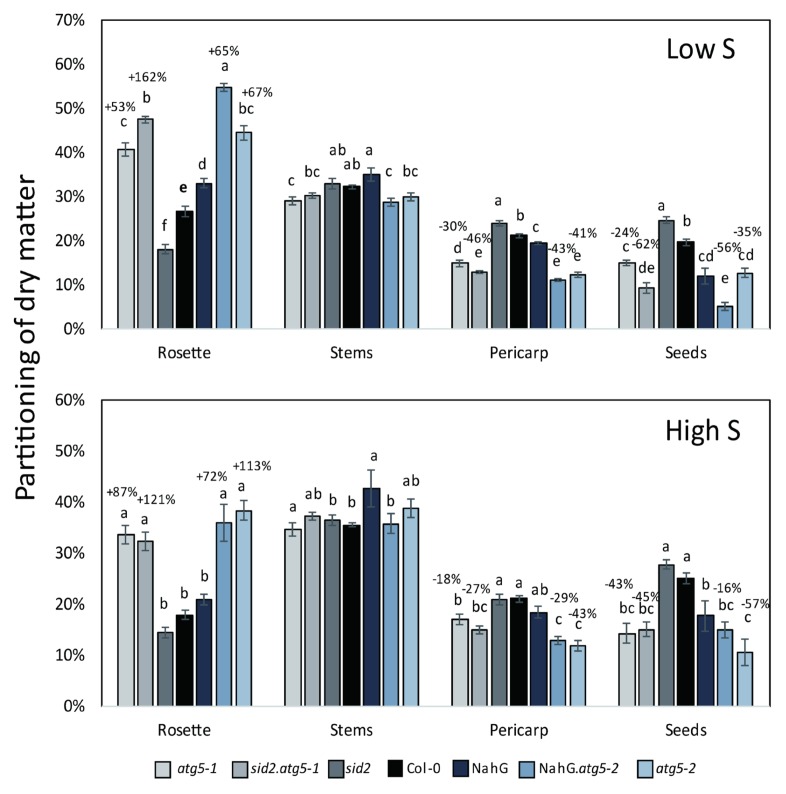
The *atg5* autophagy mutants display a lower harvest index than the control lines under both low S and high S. From the measurement of the dry weights of the rosette, stem, pericarp and seeds presented in Figure 1A, the partitioning of the dry matter in each organ was computed. The numbers above the *atg5* mutant bars indicate the % of significant increase or decrease in each mutant relative to its respective control line. Data are the adjusted means and SD from two biological repeats with six plants each. The different letters indicate values significantly different at *p* < 0.05 (n = 12), as determined using an ANOVA Newman–Keuls (SNK) comparison.

**Figure 3 cells-09-00332-f003:**
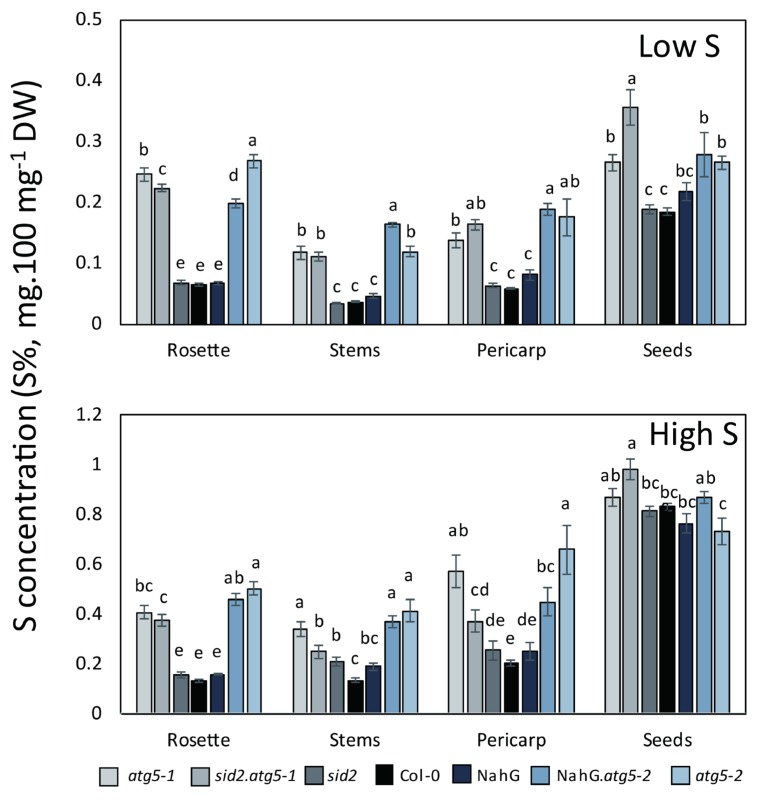
Sulphur concentrations are higher in the *atg5* autophagy mutants than in the control lines under both low S and high S. S concentrations (S% as mg.100 mg^-1^ DW) in the rosette, stem, pericarp and seeds of *atg5* mutants (*atg5-1*, *atg5-2*, *atg5-1.sid2*, *atg5-2.*NahG) and control lines (Col-0, *sid2* and NahG) were determined on plants grown under low S and high S. Data are the adjusted means and SD from two biological repeats with six plants each. The different letters indicate values significantly different at *p* < 0.05 (n = 12), as determined using an ANOVA Newman–Keuls (SNK) comparison.

**Figure 4 cells-09-00332-f004:**
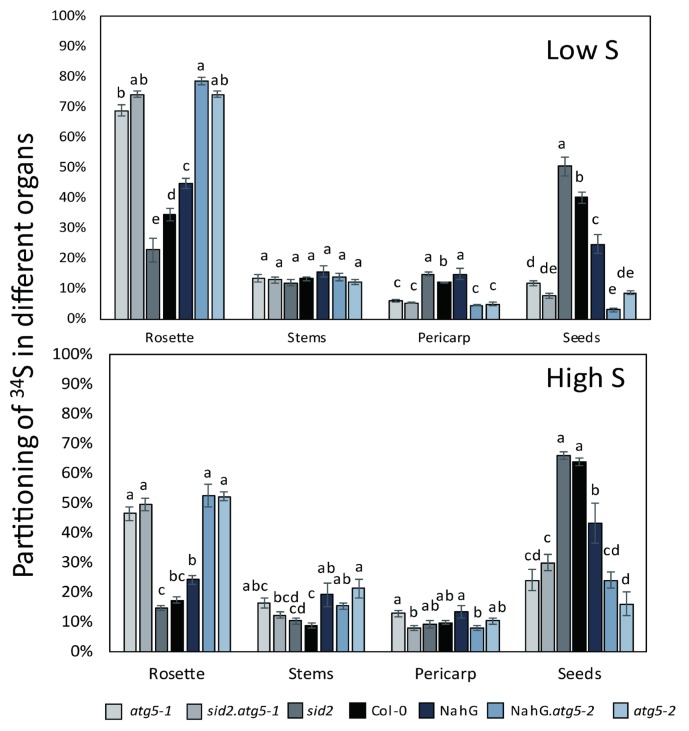
The partitioning of ^34^S in the rosette and seeds of *atg5* autophagy mutants was significantly different from that of the control lines under both low S and high S. The partitioning of ^34^S in each organ was calculated as the % of ^34^S in each organ relative to the total quantity of ^34^S in the whole plant (see material and methods and supplemental Appendix A). Data are the adjusted means and SD from two biological repeats with six plants each. The different letters indicate values significantly different at *p* < 0.05 (n = 12), as determined using an ANOVA Newman–Keuls (SNK) comparison.

**Figure 5 cells-09-00332-f005:**
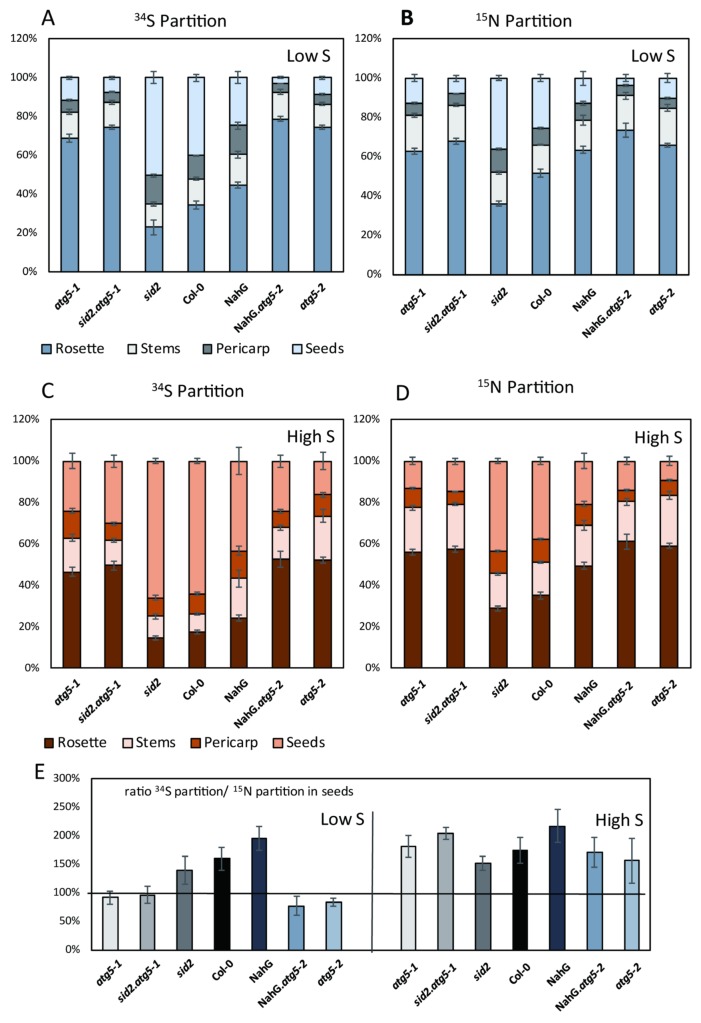
The remobilization of ^34^S and ^15^N to the seeds is differentially modified by sulphate availability in *atg5* autophagy mutants and control lines. The partitioning of ^34^S (**A,C**) and ^15^N (**B,D**) in the rosette, stem, pericarp and seeds of *atg5* mutants and control lines under low S (**A,B**) and high S (**C,D**) conditions. The ratios of ^34^S and ^15^N partitions in the seeds show different patterns depending on genotypes under high S and low S conditions (**E**). Data are the adjusted means and SD from two biological repeats with six plants each. The different letters indicate values significantly different at *p* < 0.05 (n = 12), as determined using an ANOVA Newman–Keuls (SNK) comparison.

**Figure 6 cells-09-00332-f006:**
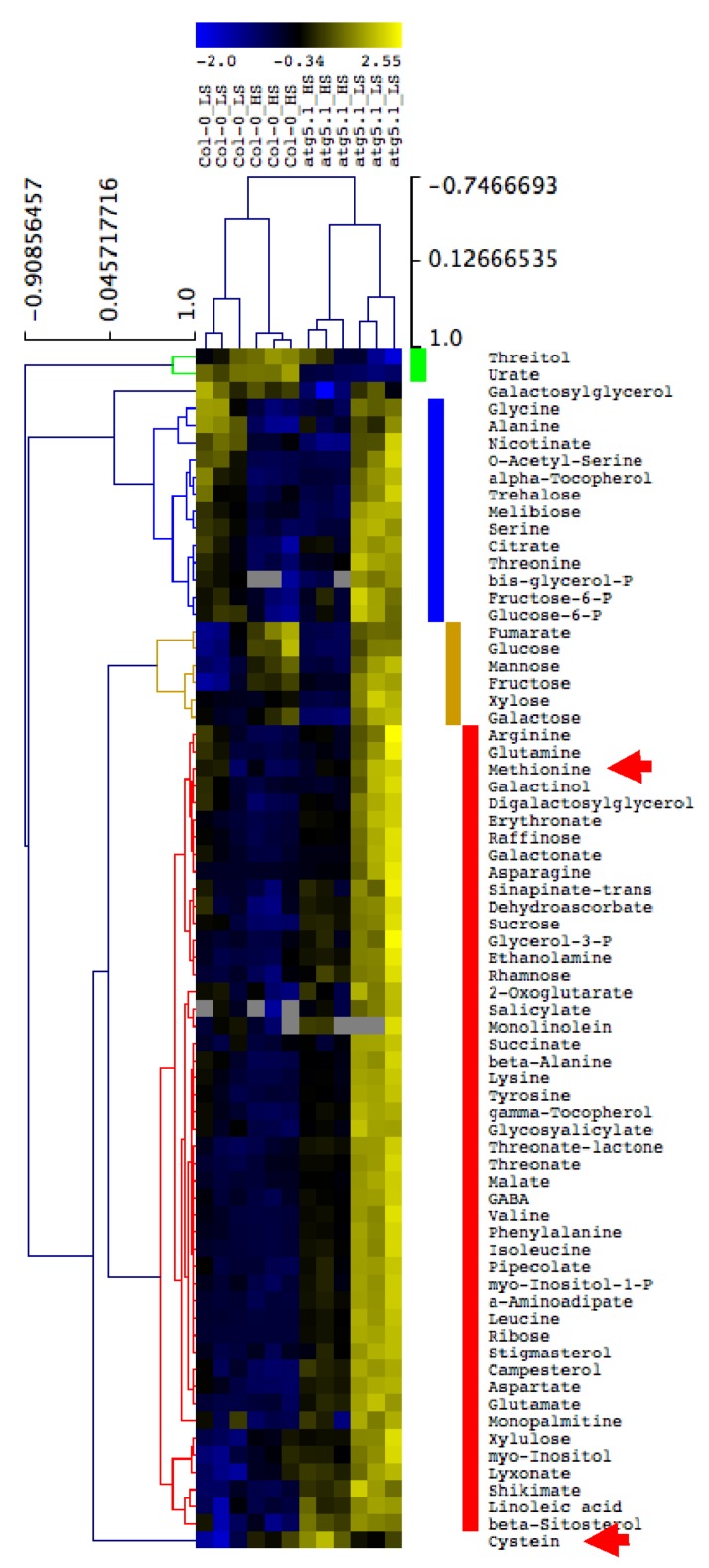
Hierarchic clustering of metabolite relative contents in *atg5-1* and Col-0. Metabolite relative contents in the rosettes of *atg5* and Col-0 wild type genotypes, grown under high sulphate and low sulphate conditions for 60 days, were measured using GC–MS. An ANOVA was performed to identify metabolites significantly modified by genotype or sulphate availability. The hierarchic clustering of significant metabolites was performed using MEV4. The clusters show metabolites less abundant in *atg5* relative to Col-0 under both high and low sulphate conditions (green), less abundant in *atg5* relative to Col-0 under high S but more abundant under low S (orange), more abundant in *atg5* relative to Col-0 under both high and low sulphate conditions (red), and not different in *atg5* and Col-0 but modified by sulphate availability (blue). The red arrows indicate the position of cysteine and methionine. Three biological repeats are shown for each genotype and condition.

**Figure 7 cells-09-00332-f007:**
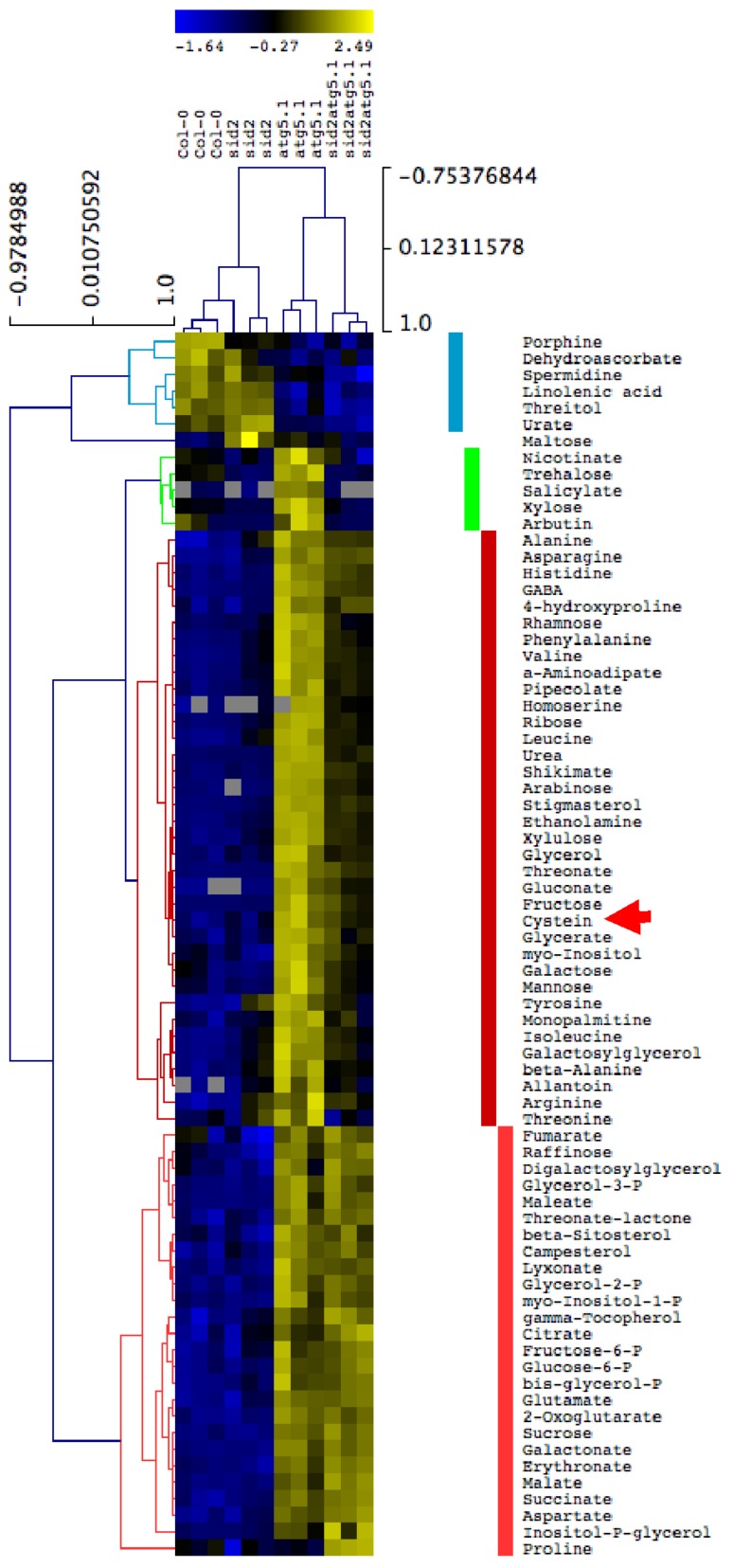
Hierarchic clustering of metabolite relative contents in *atg5-1, atg5-1.*sid2, *sid2* and Col-0 under low S conditions. The metabolite relative contents in the rosettes of *atg5, atg5-1.*sid2, *sid2* and Col-0 plants grown under low sulphate conditions for 60 days were measured using GC-MS. An ANOVA was performed to identify metabolites significantly modified by genotype. The hierarchic clustering of significant metabolites was performed MEV4. The clusters show metabolites less abundant in *atg5-1* and *atg5-1.sid2* relative to Col-0 and sid2, respectively (blue), more abundant in *atg5-1* and *atg5-1.sid2* relative to Col-0 and sid2, respectively (red), and more abundant in *atg5-1* only by comparison with the three other genotypes (green). The red arrow indicates the position of cysteine. Three biological repeats are shown for each genotype.

**Figure 8 cells-09-00332-f008:**
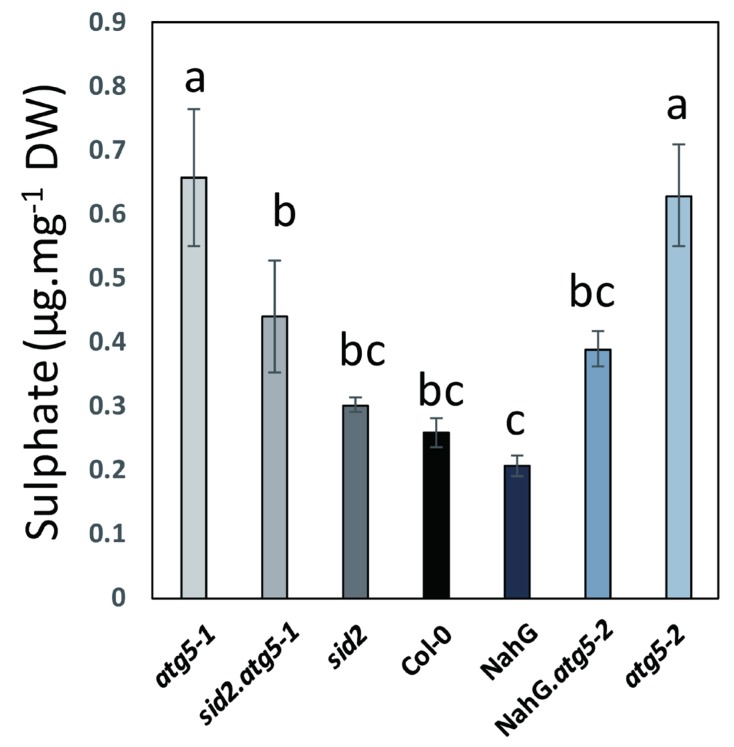
Sulphate concentrations in the rosette of *atg5* autophagy mutants are significantly higher than in the rosettes of control lines under low S. Sulphate was measured on the dry remains of the rosettes at seed maturity. Data are the adjusted means and SD from six plant repeats The different letters indicate values significantly different at *p* < 0.05 (n = 6), as determined using an ANOVA Newman–Keuls (SNK) comparison.

**Figure 9 cells-09-00332-f009:**
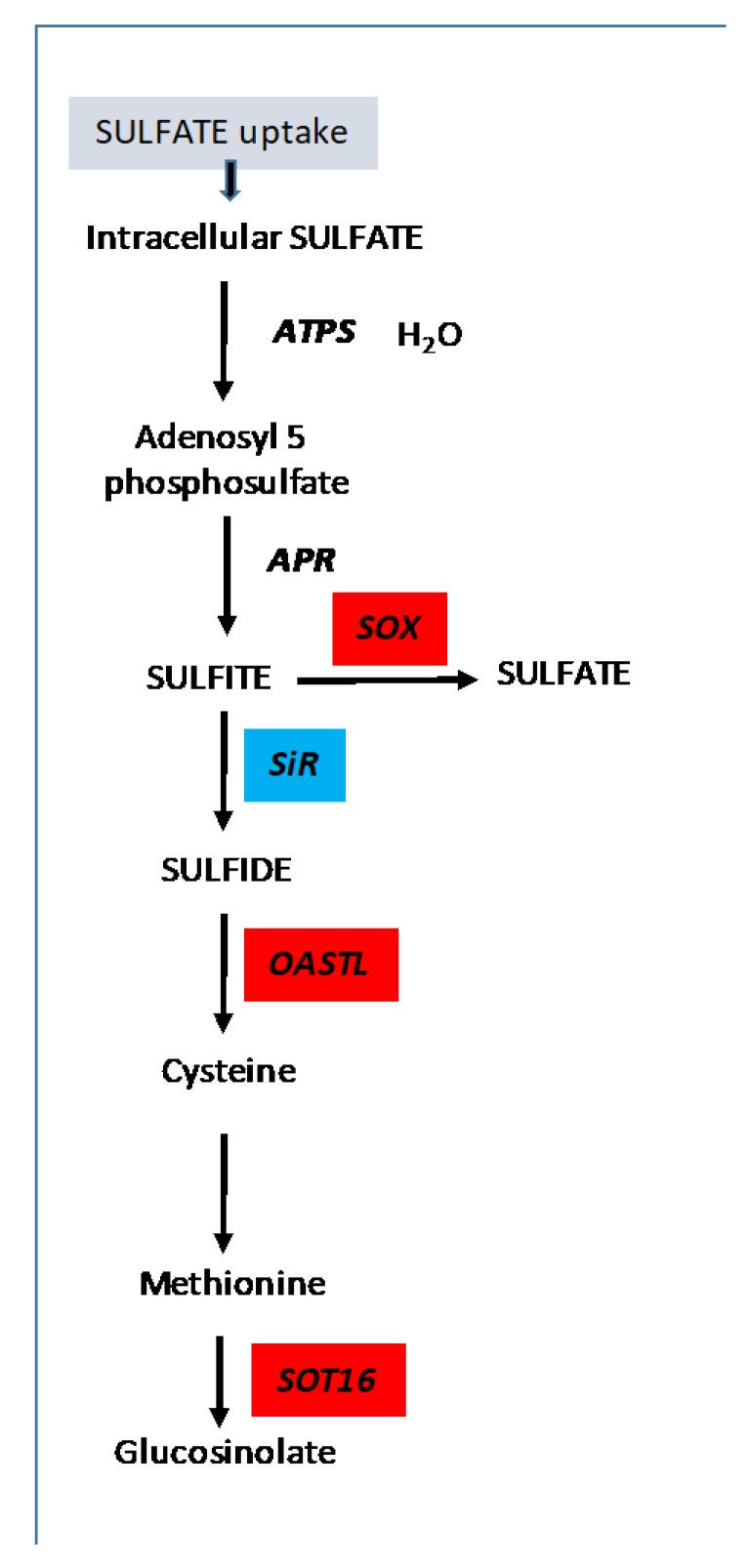
Schematic representation of the effect of *atg5* mutation on sulphur assimilation in plants grown under low S conditions. Protein accumulation or depletion in *atg5* and *atg5.sid2* mutants has been reported by Havé et al. [21]. Proteins overaccumulated in *atg5* or *atg5.sid2* vs. Col-0 or *sid2* are presented in red boxes. Proteins depleted are presented in blue boxes. ATPS, ATP sulphurylase; APR, adenosine 5-phosphosulphate reductase; SiR, sulphite reductase; OAS, O-acetylserine; OASTL, O-acetylserine (thiol) lyase; SOT16, cytosolic sulphotransferase 16.

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
