# Peer review of "Autophagy Controls Sulphur Metabolism in the Rosette Leaves of Arabidopsis and Facilitates S Remobilization to the Seeds"

_cells, 2020, doi:10.3390/cells9020332_

Round 1
Reviewer 1 Report
This article analyzes the involvement of autophagy in the Sulphur remobilization to the seeds. To do so, authors compare 34S and 15N remobilizations from the rosette to the seeds in control and atg5 mutants.
My main concern about these results is if these data are specific for atg5 mutant and if overexpressing ATG5 would increase remobilization of Sulphur. I would suggest to include those data in this publication.
Minor revisions:
-I got wrong symbols in all figures. Please check that.
-Be consistent with remobilization and remobilisation.
-Axes titles are confusing. Ex. Figure 4, portioning of 34S, is that 34S abundance calculated (A(34S)%?
-Include panel A and B from Fig.1 in the text, they are not cited.
-Statistics for panel C fig.1?
-Figure 3 and 4, Why total Sulphur in seeds is higher in atg5 than in control plants? I would expect it lower like translocation of 34S in figure 4.+
-Please keep consistency in graph legends, titles, etc. Some figures do not have them.
-Figure 3. S%=g.100 mg-1 DW, or mg.100mg-1 DW?
Author Response
My main concern about these results is if these data are specific for atg5 mutant and if overexpressing ATG5 would increase remobilization of Sulphur. I would suggest to include those data in this publication.
Response: Growing plants and labelling take almost 9 month. We cannot rule this experiment and have data ready in time. But we agree this is interesting suggestion.
Minor revisions:
-I got wrong symbols in all figures. Please check that.
Response : The strange symbols appeared when conversion to pdf was done. We need to fix that. We are sorry for inconvenience
-Be consistent with remobilization and remobilisation.
Response : We homogenised the text.
-Axes titles are confusing. Ex. Figure 4, portioning of 34S, is that 34S abundance calculated (A(34S)%?
Response : Partitioning is not isotope enrichment. We modified the legend of Figure 4 and the material and method to better explain and avoid confusion.
-Include panel A and B from Fig.1 in the text, they are not cited.
Response : These panels are now cited.
-Statistics for panel C fig.1?
Response : Statistics were added in the figure for the total biomass (rosette + stem+ pericarp+ seeeds). Legend was modified accordingly.
-Figure 3 and 4, Why total Sulphur in seeds is higher in atg5 than in control plants? I would expect it lower like translocation of 34S in figure 4.
Response : The total sulphur quantity is not presented in this paper, but we showed the S concentration (S%). Total sulphur in seeds (not shown) is not higher in atg5 than in control (under low S it was 0.24 mg in atg5 and 0.73 mg in Col for example). What is however bfascinating is that (i) despite the fact that S remobilization is decreased in atg5 mutants, the S% (S concentration) in their seeds is maintained almost as in control lines and (ii) while S% in the other organs of the atg5 mutants are much lower than in control lines, S% in seeds is maintained. The explanation is that the lower S flux to the seeds in atg5 compared to control lines has two concequences : it lowers the amount of S allocated to the reproductive organs and it results in less seed production. Indeed, the biomass of the rosette (source leaves) is 2 timess less in atg5 than in control lines at harvest irrespective of S conditions. The biomass of the seeds of atg5 is 4.5 and 6.2 times less than that of control lines under lowS and high S respectively.The S content (in mg) in atg5 is 3 and 6 times less than that of Col under lowS and high S respectively.The 34S content (in mg) in atg5 is 3.5 and 3.2 times less that of Col under lowS and high S respectively.Then we see that defects in autophagy has a stronger effect on seed production than on S allocation to the seeds. As the result, autophagy mutants produce less seeds and these seeds are properly filled with S. Similar feature was shown for N by Guiboileau et al. (2012).We added a paragraph(lines 226-240) in the text to clarify this point.
-Please keep consistency in graph legends, titles, etc. Some figures do not have them.
Response: done.
-Figure 3. S%=g.100 mg-1 DW, or mg.100mg-1 DW?
Response: Right, this is now changed in the text and figures.
Reviewer 2 Report
In this manuscript, Lornac et al., investigate the role of autophagy in sulphur metabolism and sulphur remobilization efficiency to the seeds, under both, low and high, sulphur conditions. This is a very good manuscript in many ways. Sulphur use efficiency has been not as extensively studied as the nitrogen and phosphate use efficiency, probably because previously sulfur was not limiting for agriculture. However, with the reduction of atmospheric sulfur dioxide emissions sulfur deficiency has become more common. On the other hand, autophagy is a major pathway that recycles cellular components in eukaryotic cells both under stressed and non-stressed conditions. Thus investigating the role of autophagy in sulphur remobilization is very interesting and very timely.
The results are meaningful, and the conclusions are correctly drawn.
I only have minor, editorial comments:
1. Fig.1C - please use the same font size as in Fig2.
2. Fig.3 - Description of lines is missing
3. Sulphur concentration marked as "Low S " and High S" is sometimes placed in the upper right corner or upper left corner - for easier reading please keep consistency.
Author Response
Thank you for positive comments. we took into account all of them in the revised version.
Reviewer 3 Report
In this study, the authors have compared the role of autophagy in sulphur and nitrogen management using the whole plants by performing concurrent labelling with 34S and 15N isotopes on atg5 mutants, SA-related mutants and control lines. They showed that both 34S and 15N remobilizations from the rosette to the seeds are impaired in the atg5 mutants irrespective of salicylic acid accumulation and of sulphur nutrition, in particularly under low S condition. Overall the data are well presented to support their conclusions. Below are some minor comments:
It is claimed that the defects in atg5 mutant is independent of SA level. However, in Figure 8. It seems that the Sulphate concentrations in the rosette of atg5/sid2, atg5/nahG mutants are significantly higher than in the rosettes of control lines under Low S, but less than atg5 single mutant. This suggests that SA level will influence the effect of the autophagy defect on the Sulphate metabolism, at least partially. Therefore, the conclusion need to be revised. “Under low S, it is likely that the absence of autophagy mainly affects the transport of N-poor S-364 containing molecules.” Does all the changes of Sulphate concentrations only depend on the transport rate of the metabolic molecules? Is it possible that the degradation process of S-containing molecules (proteins/lipids) is blocked to different levels in different tissues (the role of autophagy)?
Author Response
Response. We thank reviewer for his/her positive comments.
Conclusion was modified taking into account the SA-dependent effect (lines 425-426).
Sentences and references relative to the role of autophagy in lipid degradation were added (lines 396-400).